# Schistosomiasis screening in non-endemic countries from a cost perspective: Knowledge gaps and research priorities. The case of African long-term residents in a Metropolitan Area, Spain

**Sílvia Roure[1,2,3], Francesc López[4,5,6], Irene Oliva[5], Olga Pérez-Quílez[1], Oriol March[5], Anna Chamorro[2], Elena Abad[2], Israel López Muñoz[1], Amaia Castillo[1], Laura Soldevila[1,3], Lluís Valerio[1], Manolo Lozano[7], Helena Masnou[8], Mario Oliveira[9], Laura Cañas[10], Mireia Gibrat[11], Marta Chuecos[12], Juan José Montero[12], Karen Colmenares[13], Gemma Falguera[14], Josep Maria Bonet[14], Mar Isnard[14], Núria Prat[14], Oriol Estrada[5,15], Bonaventura Clotet[2,16,17], Xavier Vallès[1,2,18] ***

**1** International Health Program (PROSICS), Direcció Territorial de Malalties Infeccioses Metropolitana Nord, Institut Català de la Salut, Badalona, Spain, **2** Fight and Infectious Diseases Foundation, Badalona, Spain, **3** Infectious Diseases Unit, Hospital Universitari Germans Trias i Pujol, Badalona, Spain, **4** Centre de Recerca en Economia de la Salut (CRES), Universitat Pompeu Fabra, Barcelona, Spain, **5** Grup de Recerca en Innovació, Economia de la Salut i Transformació Digital (Institut de Recerca Germans Trias i Pujol–IGTP), Badalona, Spain, **6** Gerència Territorial Metropolitana Nord, Institut Català de la Salut, Badalona, Spain, **7** Neurology Department, Hospital Universitari Germans Trias i Pujol, Badalona, Spain, **8** Gastroenterology Department, Hospital Universitari Germans Trias i Pujol, Badalona, Spain, **9** Urology Department, Hospital Universitari Germans Trias i Pujol, Badalona, Spain, **10** Nephrology Department, Hospital Universitari Germans Trias i Pujol, Badalona, Spain, **11** Primary Health Care Unit Canovelles, North Metropolitan Health Region from Barcelona, Institut Català de la Salut, Granollers, Spain, **12** Primary Health Care Unit Mataró-3 (Rocafonda-Palau), North Metropolitan Health Region from Barcelona, Institut Català de la Salut, Badalona, Spain, **13** Unitat de Suport Assistencial i Avaluació (USUAiA), Direcció d'Atenció Primària Metropolitana Nord, Institut Català de la Salut, Sabadell, Spain, **14** North Metropolitan Primary Care Directorate, Instiut Català de la Salut, Sabadell, Spain, **15** Directorate for Innovation and Interdisciplinary Cooperation, North Metropolitan Territorial Health Region, Institut Català de la Salut, Badalona, Spain, **16** IrsiCaixa—Institut de Recerca de La SIDA, Hospital Universitari Germans Trias I Pujol, Badalona, Spain, **17** Direcció Territorial Malalties Infeccioses, North Metropolitan Territorial Health Region, Institut Català de la Salut, Badalona, Spain, **18** Institut de Recerca Germans Trias i Pujol, Badalona, Spain

* xvallesc.mn.ics@gencat.cat

## Abstract

### Background

Imported schistosomiasis is an emerging issue in European countries as a result of growing global migration from schistosomiasis-endemic countries, mainly in sub-Saharan Africa. Undetected infection may lead to serious long-term complications with an associated high cost for public healthcare systems especially among long-term migrants.

### Objective

To evaluate from a health economics perspective the introduction of *schistosomiasis* screening programs in non-endemic countries with high prevalence of long-term migrants.

**Data Availability Statement:** All relevant data are within the paper and its Supporting Information files.

**Funding:** The authors received no specific funding for this work.

**Competing interests:** The authors have declared that no competing interests exist.

## Methodology

We calculated the costs associated with three approaches—presumptive treatment, test-and-treat and watchful waiting—under different scenarios of prevalence, treatment efficacy and the cost of care resulting from long-term morbidity. Costs were estimated for our study area, in which there are reported to reside 74,000 individuals who have been exposed to the infection. Additionally, we methodically reviewed the potential factors that could affect the cost/benefit ratio of a schistosomiasis screening program and need therefore to be ascertained.

## Results

Assuming a 24% prevalence of schistosomiasis in the exposed population and 100% treatment efficacy, the estimated associated cost per infected person of a watchful waiting strategy would be €2,424, that of a presumptive treatment strategy would be €970 and that of a test-and-treat strategy would be €360. The difference in averted costs between test-and-treat and watchful waiting strategies ranges from nearly €60 million in scenarios of high prevalence and treatment efficacy, to a neutral costs ratio when these parameters are halved. However, there are important gaps in our understanding of issues such as the efficacy of treatment in infected long-term residents, the natural history of schistosomiasis in long-term migrants and the feasibility of screening programs.

## Conclusion

Our results support the roll-out of a schistosomiasis screening program based on a test-and-treat strategy from a health economics perspective under the most likely projected scenarios, but important knowledge gaps should be addressed for a more accurate estimations among long-term migrants.

## Author summary

At present, screening for schistosomiasis among long-term migrants to non-endemic countries is relatively uncommon, despite growing evidence of the burden to healthcare systems associated with chronic disease. In this article we estimate the costs of systematically screening an exposed population for schistosomiasis infection. Though our results support the implementation of such a program from a cost perspective, they are hampered by important gaps in our ability to estimate costs, particularly with regard to the efficacy of treatment of chronic *Schistosoma* infection in adults. Therefore the implementation of any screening program should be aligned with further research regarding these costs. Screening programs would also benefit from the development of in-situ diagnostic tests and an appropriate Point-of-Care strategy.

## Introduction

Schistosomiasis is a parasitic infection that affects 229 million people worldwide, mainly in sub-Saharan Africa, where it accounts for most of the annual incidence (>90%), morbidity and mortality. In 2017, the global burden of schistosomiasis was estimated at 1.4 million

disability-adjusted life-years [1, 2]. In spite of these numbers, it is still regarded by the World Health Organization as a neglected disease. Its clinical manifestations depend mostly on the organ or tissue it targets. While respective targets are the genitourinary tract for *Schistosoma haematobium* and digestive tract for *Schistosoma mansoni*, the two most prevalent species, other body systems may be affected. The impact of four other species, *Schistosoma intercalatum*, *Schistosoma japonicum*, *Schistosoma mekongi* and *Schistosoma guineensis*, is much more limited [3]. The presence in the human body of *Schistosoma* worms and eggs, which can remain there for up to 30 years, induces an immune-mediated granulomatous response that causes local and systemic pathological effects which may lead to long-term serious complications such as chronic renal failure, pulmonary hypertension, liver cirrhosis and stroke [3–5].

The current treatment of choice for schistosomiasis infection, Praziquantel (PZQ), is safe, low-cost and highly effective. If administered in time, PZQ can avert long-term and irreversible chronic complications [3,5,6]. The impact of schistosomiasis on public health systems in non-endemic regions like Europe and North America has risen lately due to increasing global migration from endemic regions, mostly sub-Saharan Africa [7, 8]. The most comprehensive estimates indicate that the sero-prevalence of schistosomiasis among migrants from high-endemic countries who are living in non-endemic countries may be as high as 24.1% (CI 16.4–32.7) [9]. However, Schistosoma screening programs are scarce in Europe and clinical suspicion of the infection very low, in spite of the recommendation by the ECDC to do so in all arriving migrants [10] and the evidences collected so far [11]. Late diagnostic of schistosomiasis infection and associated morbidities seems to be the rule [11–13]. This means that many cases may pass undetected, leading to the development of long-term chronic complications with concomitant high care-associated costs (e.g. dialysis for patients with terminal renal insufficiency) [11–14]. The main issue about laboratory methods is the lack of specificity towards active infection (serology-based kits) and lack of sensitivity of urine/stool examination in non-endemic countries [15, 16]. Overall, this may lead to an unknown prevalence of Schistosomiasis infection and associated morbidity especially among long-term resident migrants. Therefore, various strategies have been proposed to deal with this emerging public health issue, such as presumptive mass treatment of exposed populations or serological and clinical screening [13] based on passive or active case detection or the combination of the two-step screening process with serology and stool or urine examination [17]. Regardless of the particular strategy adopted, however, given the scale of the problem, any screening and prevention program that is ultimately implemented will need to exhibit a high degree of cost-effectiveness.

Our study aims to raise the gaps in current knowledge regarding a variety of issues impacting the cost of schistosomiasis screening program in non-endemic countries. Based on these observations, we build a simple static cost analysis comparing a watchful waiting approach, currently the status quo in most non-endemic countries, with either presumptive treatment or test-and-treat strategies, considering different reasonable scenarios and necessary assumptions. Finally, the analysis allows us to point out the areas where further research is necessary.

## Material and methods

This is a four steps analysis. At first instance, in order to identify factors that might influence the costs and costs/benefit ratio of a schistosomiasis screening program, we hosted an appraisal that included health professionals working in the area of imported diseases and/or migrant populations and health economists in our study area. For each such factor identified, we then carried out a narrative review of the current literature. We determined the realistic bounds of factors which could reasonably be estimated among our study population (i.e. prevalence of infection) to be applied to our model. At second instance, we develop a modelling economic

study with a static approach based on our study population, considering different scenarios and assumptions according to the previous assessment, comparing three strategies (wait and watchful, presumptive treatment and test and treat). Thirdly, we confronted the model with additional identified co-factors from which we could not yet ascertain reasonable bounds estimations and should be subject to further research or evaluation (see results section). Finally, we develop a list of research priorities raised by our analysis, in order to determine or develop the most suitable schistosomiasis-screening strategy of imported schistosomiasis in non-endemic countries (see discussion section).

## 1. Main factors that might influence to the costs associated with Screening strategies

i.  *Schistosomiasis prevalence.* The most comprehensive study estimates the average sero-prevalence of schistosomiasis infection among sub-Saharan African migrants in Europe to be 24.1% [9]. The 56.8% sero-prevalence observed in primary care wards in the area covered by the present study [13] may exceed actual community sero-prevalence since it may over-represent only the symptomatic sub-set of the population. However, the correlation between serology (sero-prevalence) and active infection susceptible to respond to specific treatment (real prevalence) is far from clear, specifically among long-term migrants. In our calculations, we therefore considered a band of schistosomiasis prevalence scenarios ranging from 5% to 50%.

ii.  *Associated care costs.* Not all morbidity-associated care procedures are indicated for any given patient, and individual-level costs may vary substantially, including the stage when a long-term complication has been detected (e.g. simple renal insufficiency vs. terminal renal insufficiency). Consequently, we considered different scenarios for associated care costs between 40% of the maximum estimated cost to 100% of those displayed in Table A in S1 Text

iii.  *Treatment efficacy.* PZQ efficacy has been tested against proven infection, which normally means the observation of eggs in urine or stool. However, this is the exception in non-endemic countries or long-term infections. The efficacy of treatment after years since last exposure remains unknown (see next section). Therefore, we considered different reasonable scenarios of average efficacy of treatment ranging from 40% to 100%. Even though a 100% cure is considered as a reference point, not a realistic expectative.

## 2. Cost-effectiveness model build-up

**Potential beneficiaries of schistosomiasis screening.**   For the purposes of this study we regarded as schistosomiasis-exposed any individual who had spent a significant period of their life in a schistosomiasis-endemic region and was living in the study catchment region (north metropolitan area of Barcelona, Spain) at the time of the study. By this definition, the official census estimated that around 74,000 individuals living in the study area qualified as schistosomiasis-exposed [18], most of them migrants from sub-Saharan Africa, of which 28% were females (N = 20,720) and 72% males (N = 53,280) [18]. The mean age of migrant population in our study area is 34.3 yrs. substantially younger than locals (44,4 yrs.) [18] and 32.7% come from Senegal (see the list of main countries of origin in Table 1). According to our own records 63% have been living in our country for more than 10 years [13]. Assuming a *Schistosoma* infection prevalence of 24%, the total number of *Schistosoma*-infected individuals could be as high as 17,800. This prevalence is consistent with that seen in the six schistosomiasis-endemic countries most heavily represented in the local immigrant community (see Table 1).

**Table 1. Estimated population-adjusted prevalence of schistosomiasis in sub-Saharan African migrant communities living in Catalonia, Spain, in 2012[19] [1].**

| Country | % of local sub-Saharan migrant population | Prevalence S. haematobium | | Prevalence S. mansoni | | Overall Schistosoma prevalence | |
|---|---|---|---|---|---|---|---|
| | | % | CI | % | CI | % | CI |
| Senegal | 32.7 | 18.3 | (16.2–21.3) | 2.2 | (1.5–4.1) | 20.3 | (17.9–23.5) |
| Gambia | 20.7 | 16.8 | (10.4–32.3) | 7.5 | (1.9–23.8) | 24.5 | (15.1–43.0) |
| Mali | 10.7 | 29.4 | (26.6–32.5) | 7.8 | (6.2–9.5) | 34.2 | (31.3–37.1) |
| Ghana | 9.7 | 22.3 | (19.4–26.5) | 1.3 | (0.7–2.3) | 23.3 | (20.5–27.3) |
| Nigeria | 8.8 | 22.0 | (19.9–24.6) | 4.4 | (3.2–6.0) | 25.2 | (23.0–27.8) |
| Guinea | 5.8 | 12.3 | (8.3–17.1) | 14.6 | (9,9–20,5) | 24.4 | (20.5–27.3) |

[1]These estimations correspond to the observation of eggs in urine or stool samples and mostly coming from children or young adults, whereas the estimations in Europe are from sero-prevalence studies. The real prevalence of the active infection in long-term migrants in Europe remains unknown.

For our cost analysis we considered different community-based prevalence projections ranging from a low prevalence of 5% to a high one of 50%, assuming an equal probability of infection between genders. We regarded the 74,000 exposed individuals as a static cohort. Costs were assessed in a life-time perspective.

**Approaches to schistosomiasis control.** The three strategies we compared were: 1) a watchful waiting strategy (WW), whereby no screening and/or preventive measures would be taken, with *Schistosoma*-associated pathologies treated upon diagnosis or manifestation; 2) a presumptive treatment strategy (PT), whereby all exposed individuals would be treated with PZQ without any prior screening; and 3) a test-and-treat strategy (TT), whereby a serology-based test would be carried out on all *Schistosoma*-exposed individuals, with all those testing positive being subsequently treated with a standard dosage of PZQ. For approaches 2 and 3 we assumed that the workload involved would be assumed by the primary care services, whereas long-term manifestations would be dealt with at hospital level.

**Cost analysis for each strategy.**

I. *Costs associated with a WW strategy*

The estimated cost of the WW strategy was obtained through the weighted sum of the costs of each pathology by its probability of occurrence in each individual infected by schistosomiasis according to the reference literature [20, 21]. We considered the eighteen most frequent *Schistosoma*-associated clinical manifestations and morbidities (see **Table 2**). The resources required for care and treatment of each of these pathologies were identified by healthcare professionals and the corresponding costs were obtained from the healthcare provider 2020 tariffs (see Table A in S1 text) [22]. In other words, this estimate represents the maximum amount of resources which could potentially be averted if the disease were detected in time, the symptoms treated before the long-term complications appeared and under the theoretical assumption of 100% effectiveness of treatment (see below for more details). Costs of gender-specific morbidities were counted only for the corresponding gender category (i.e. female genital schistosomiasis and infertility among females). Costs were estimated from the health system perspective and assuming a lifetime time horizon.

II. *Costs associated with PT and TT strategies*

The costs associated with a primary care visit and a serological test were €50 and €24 respectively (which includes sample collection, transport, laboratory procedures, material and reagents and personnel). We assumed one visit for each screened or presumptively treated

**Table 2. Long-term schistosomiasis-associated pathologies and associated care procedures.**

| Pathology | Associated resources |
| --- | --- |
| Hepatomegaly | Hepatopathy study, abdominal ultrasound, first visit, liver elastography, follow-up visit (×2). |
| Blood in the stool | Analysis with hemogram and anemia study, colonoscopy, first visit, follow-up visit (×2). |
| Splenomegaly | Hepatopathy study, abdominal ultrasound, first visit, follow-up visit (×2). |
| Ascites | Hepatopathy study, abdominal ultrasound, diagnostic paracentesis, biochemistry and cytology, 24-hour urine, biochemistry, sediment + urine cultivation, biopsy, transjugular liver exploration, hospital admission, fibrogastroscopy, first visit, follow-up visit (×2). |
| Hematemesis | Fibrogastroscopy, esophageal varices ligation, hospital admission (from 1st to 6th), transjugular liver biopsy, Intravenous iron or blood transfusion, hepatopathy study, liver elastography, abdominal ultrasound, first visit, follow-up visit (×2). |
| Hematuria | First visit, follow-up visit, urinary system ultrasound, sediment + urine cultivation, serial urine cytology (3 samples), transurethral cystoscopy. |
| Bladder cancer | TUR of bladder tumor, cystoscopy, urine cytology every 3 months, CT urography, abdominal thoracic CT, radical cystectomy, cutaneous ureteroileostomy (Bricker), Thoracic lymphadenectomy, first visit, follow-up visit (×2), CT at 3 months and every six months. |
| Dysuria/minor bladder morbidity | First visit, follow-up visit (×2), urinary system ultrasound, sediment + urine cultivation, cystoscopy, serial urine cytology (3 samples). |
| Major bladder morbidity | First visit, follow-up visit (×2), urinary system ultrasound, Sediment + urine cultivation, Transurethral cystoscopy, Serial urine cytology (3 samples), basic urodynamics, uroflowmetry. |
| Hydronephrosis/ urethral stenosis | First visit, follow-up visit, urinary system ultrasound, urine sediment + urine culture, CT urography, MAG3 renogram with furosemide. |
| Severe kidney failure | Laboratory tests with coagulation and serologies, abdominal ultrasound, ultrasound-guided biopsy, AVF for dialysis, central venous catheter tunneled through the jugular vein (DIVAS), chest X-ray, abdominal CT angiography, catheter administration for dialysis, echocardiogram, dialysis (1 session), 3 sets/week until death/transplant, liver transplant. |
| Female infertility | First visit, follow-up visit, transvaginal ultrasound, in vitro fertilization (IVF), IVF cycle, frozen embryo transfer (FET), hysterosalpingography. |
| Stroke | Cranial CT with CT angiography, cranial MRI, laboratory tests with coagulation, echocardiogram, transcranial Doppler ultrasound, electrocardiogram (ECG), hospital admission (X = 8 days), rehabilitation, first visit, follow-up visit × (2). |
| Myelitis | Spinal resonance, lumbar puncture, spinal arteriography, vesical catheterization, hospital admission (1st to 6th) 15 days, first visit, follow-up visit (×2). |
| Female genital schistosomiasis | First visit, follow-up visit, cytology, transvaginal ultrasound, cervical biopsy. |
| Glomerulonephritis | First visit, analysis with proteinuria, abdominal ultrasound, renal Doppler, ultrasound-guided kidney biopsy, analysis of the biopsy for anatomical pathology, follow-up visit every 4–5 months. |
| Pulmonary hypertension | First visit, follow-up visit, transthoracic ECG, ECG, Chest X-ray, high resolution thoracic CT, scintigraphy V/Q, spirometry+DLCO, right coronary cardiac catheterization. |

Source: Hospital Universitari Germans Trias i Pujol (see data availability statement)

individual and a second visit for a diagnosed and treated individual. The cost of treatment with PZQ was included in the estimated costs for both PT and TT strategies but included all exposed individuals for the PT strategy and only the infected (serology-positive) individuals for the TT strategy. According to CDC and WHO guidelines, one dose of PZQ is sufficient for treatment in endemic countries [23, 24]. However, at present, local protocols require the administration of a double dose. Therefore, in our analysis we considered three scenarios: a

PT strategy with one dose per person (a TT strategy with one dose per person (TT-1) and a TT strategy with a double dose per person (TT-2). In short, the PT strategy involved one visit for each exposed individual (cost per person €50) and one dose of PZQ (€80, total €130); the TT-1 involved a first visit (€50) with serological test (€24, total €74) for all exposed individuals and a second visit (€50) and single PZQ treatment (€80, total €130) for those who tested positive; and finally the TT-2 involved a visit with test (€74) for all exposed individuals and a second visit and double dose (€210) for those testing positive.

**Cost analysis outputs.** Outputs were considered as i) costs per infected person; ii) total cost for each strategy according to different prevalence scenarios (5% to 50%) and different treatment costs for long-term morbidities for the WW strategy (from 40% to 100%); and iii) the total savings margin (costs averted) between different strategies and different scenarios of *Schistosoma* prevalence (15% to 30%), schistosomiasis treatment efficacy (40% to 100%) and costs associated with long-term morbidities (40% to 100%). The model was analyzed and set-up with R version 4.1.2.

## Results

### Cost estimates

The cost estimates for the four strategies (WW, PT, TT-1 and TT-2) under different scenarios are shown in Fig 1 (total costs) and Fig 2 (average costs per infected person). Briefly, the expected cost of applying a WW strategy for each *Schistosoma*-infected individual ranged between €2,424 assuming full costs of long-term care (100%) and €970 assuming 40% of those costs, and assuming a 24% prevalence of schistosomiasis in the exposed population. This means that the total cost of applying a WW strategy assuming a life-time horizon for the

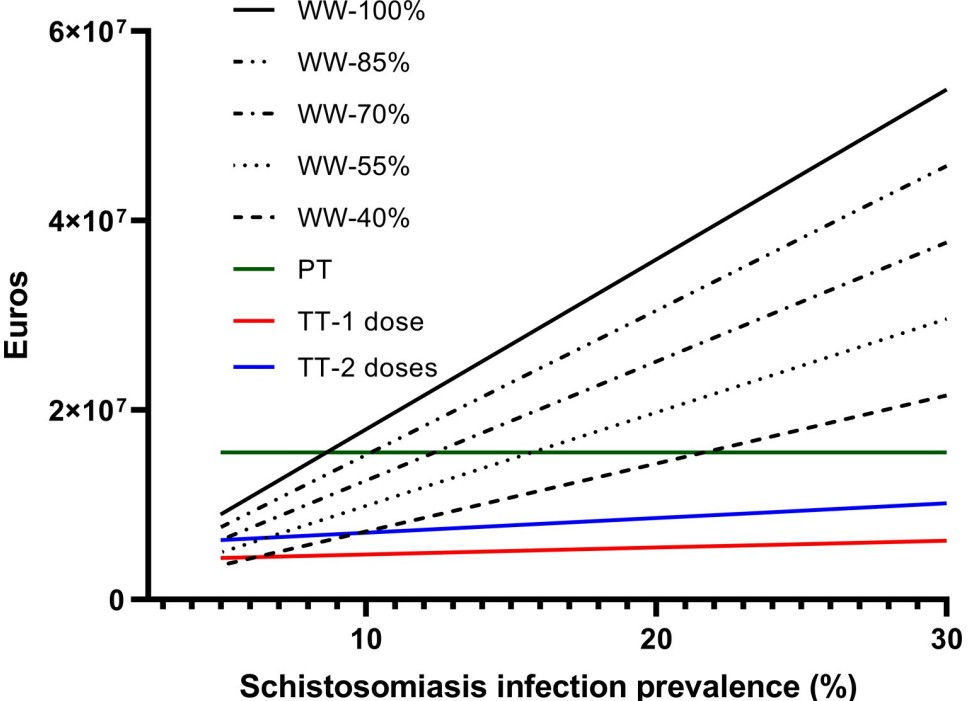

**Fig 1. Total projected cost for each strategy at different prevalence rates and for the WW strategy, at five percentages of the maximum cost of treatment of associated morbidities\*.** *WW: Watchful and waiting strategy; PT: Presumptive treatment; TT: Test and treat strategy.

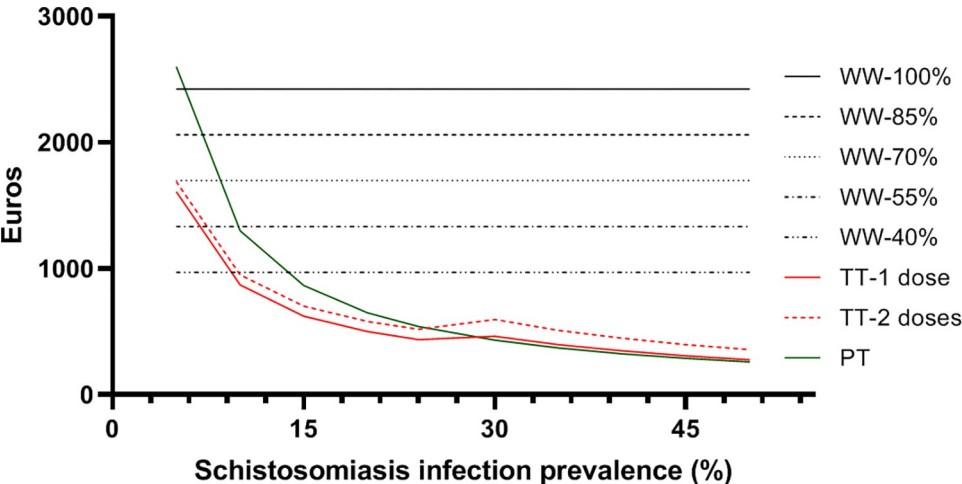

**Fig 2. Projected cost per infected individual for each strategy at different prevalence rates and for the WW strategy, at five percentages of the maximum cost of treatment of associated morbidities**\*. \*WW: Watchful and waiting strategy; PT: Presumptive treatment; TT: Test and treat strategy.

current cohort of estimated-infected individuals in the study area would range from €43 million for a 100%-costs scenarios to around €17 million for a 40%-costs scenario with a prevalence of 24%. Under a 5% prevalence assumption (lowest bound), the range would be between around € 9 and €3.5 million, respectively. As shown in Fig 1, the projected cost of a TT-2 strategy would equal that of a WW strategy at a lower prevalence scenario (10% for 40% of morbidity-associated costs) with a total of €7 million costs. With regard to the PT strategy, we observed a neutral costs at a prevalence of 12% with a total of €7 million costs. The PT strategy would match the two-dose TT-2 strategy at prevalence around 25% (see Fig 1). Considering the costs-per-infected person the cost equivalence of PT, TT-2 and TT-1 with the WW strategy at 40% of costs (970€) are estimated at prevalence of 14%, 10% and 9%, respectively (see Fig 2).

## Costs-averted analysis

The net estimated savings when applying the TT and PT strategies with respect to the WW strategy under different prevalence, treatment efficacy and treatment cost assumptions are shown in Tables B to D and Fig A in S1 Text. It will be seen that in almost all scenarios considered the TT strategy implies net savings relative to the WW strategy, which rise to €58.6 million assuming 30% prevalence in our study area, 100% treatment efficacy (admittedly unexpected) and 100% potentially averted costs (upper estimate bounds). The neutral balance was estimated at 15% prevalence, 50% of treatment efficacy and around 50% of estimated morbidities-associated costs.

## Additional co-factors that may influence the costs and costs ratio balance

During the appraisal discussions, we identified 8 additional factors which could influence the health economics of our analysis and therefore deserve further consideration:

1. *Relative prevalence of Schistosoma species infection.* Given the marked tissue tropism of each *Schistosoma* species, their incidence and the prevalence of the various long-term morbidities can differ substantially (i.e. urogenital involvement for *S. haematobium* vs. liver involvement for *S. mansoni*) [3–5]. This in turn may affect the economic analysis results by increasing or decreasing the associated care costs, which can vary greatly depending on the

specific pathology (see Table 1). Overall prevalence in sub-Saharan Africa have been estimated at 14.6% (CI 13.7–15.5) and 4.6% (4.2–5.1) for S. *haematobium* and *S. mansoni* respectively, with a total schistosomiasis prevalence estimated at 24.1% (CI 23.0–25.1) in 2012 [19]. In some countries the prevalence of *S. haematobium* can be from two to nine times higher than that of *S.mansoni*. However, in some countries (e.g. the Democratic Republic of Congo, Uganda, Guinea and the Central African Republic), the prevalence of *S. mansoni* exceeds that of *S. haematobium* [19]. Therefore, the distribution of countries of origin of migrant population and probably socio-demographic characteristics associated with higher risk of acquisition may influence the prevalence of each species in different migrant population pockets (see Table 1).

2.  *Treatment regimen and dosage.* PZQ treatment efficacy is normally reported as Cure Rate, which compares the number of pre-treatment egg-positive individuals who become negative post-PZQ treatment to those who do not, as well as by the egg reduction rate, which is determined by the reduction in mean number of eggs excreted in urine from pre-PZQ compared to post-treatment [24]. PZQ efficacy has been tested mainly in endemic countries in school-aged children with *Schistosoma* eggs observed in urine or stool, and therefore with recent and/or acute symptomatic infection [25–27]. However, the excretion of eggs in long-term exposed adult patients (like most infected migrants) is rarely observed in non-endemic countries and thus PZQ efficacy cannot be evaluated in these terms. As a result, though the current guidelines in our study area advise the administration of PZQ to any suspected infection based on serology test [10], the baseline evidence of efficacy is still lacking specially among long-term infections [28]. This includes PZQ treatment's potential rate of prevention of long-term complications. This rate could also be modified by the coinfection that frequently occurs among migrants and that worsens the prognosis for complications like VHB and VHC [29]. The optimal treatment efficacy cut-offs are still a matter of debate, including dosage regimen, and multicenter clinical trials in non-endemic areas are still needed [30], but it seems clear that a single dose does not reach 100% efficacy in the chronic phase [28]. It is also not yet clear how PZQ administration impacts chronic lesions (i.e. eggs-deposit induced fibrosis); according to some reports bladder lesions and fibrosis reverses after therapy, but advanced pathology may persist for a long time in spite of treatment or non-re-exposure [31, 32]. A previous study in our area showed an improvement in clinical and analytical abnormalities following treatment of long-term resident migrants [13]. It is reasonable to assume a certain decrease in efficacy of PZQ treatment over time after exposure and, in turn, a decrease in the cost/benefit ratio of a PT or TT program in populations that left the country of initial exposure long ago. In any case, the 100% cure is unexpected in all long-term exposed individuals. Consequently, the time elapsed since migration occurred could be a major factor influencing the costs averted by PT or TT. These observations point out first, that any screening program should include the active referral and linkage to care of patients with particular clinical manifestations for adequate follow-up, and second, that the program should aim to reach recently arrived migrants where higher cost/benefit and treatment efficacy would be expected.

3.  *Coverage of screening strategies.* The coverage of screening programs of any sort, including infectious diseases, that target migrant populations have been consistently shown to be suboptimal in Europe [33], for diverse reasons [34, 35]. A *Schistosoma* screening program would be no exception [11]. This limited coverage would diminish and thus absolute savings. Indeed, as in other infectious diseases, the hard-to-reach pockets of population are likely to be those with the highest risks of infections [36–38]; indeed, undocumented migrants residing in our study area were not considered in our analysis, since they are not

registered in official databases. Consequently, the implementation of a PT or TT strategy would ideally require the allocation of resources for active-case detection and outreach activities, with the provision of the necessary procedures recommended for health promotion programs in Europe, such as voluntary, confidential and non-stigmatizing screening and efficient linkage to appropriate care and treatment [39] as well as awareness-raising programs to increase the acceptability of screening and treatment among targeted communities. Overall, this would increase the associated costs of the PT and TT strategies, as would a feasibility analysis to achieve an acceptable degree of coverage.

4. *Limited specificity of serology-based screening.* Serology-based screening is considered the most appropriate diagnostic test in non-endemic countries [40]. However, current serology tests do not differentiate between present/active infection and past and resolved infection. It is assumed that patients with past or resolved infection tend to produce negative results, but the timing has not been clearly established. This lack of specificity in schistosomiasis serology testing can result in unnecessary treatments and hence an excess in associated costs for the TT strategy. Otherwise, a serology negative results could not roll out the existence of infection and consequently, the clinical evaluation of those patients with signs and symptoms compatible with Schistosoma infection [13].

5. *Secondary health-service-associated costs.* The cost analysis carried out here focused on the costs of the final health outcomes of an undiagnosed *Schistosoma* infection. However, symptomatic patients tend to be visited several times before a serious complication emerges or *Schistosoma* infection is finally suspected, as has been observed in our study area for female genital schistosomiasis [41]. This may represent a substantial burden on health-system resources, which would increase the averted costs of an early treatment strategy, whether TT or PT.

6. *Socio-demographic changes in the exposed migrant population.* The probability of developing long-term schistosomiasis-associated morbidities has been calculated on the basis of estimates for endemic countries. However, the life expectancy of a migrant population generally increases dramatically upon arrival in Western countries, thanks to better access to healthcare services (especially infant and maternity care), food security and the absence of high-burden endemic diseases such as malaria. By way of reference, in 2020 overall life expectancy was estimated at 62 years in sub-Saharan Africa (albeit with sharp differences between and within countries) [42], and 80.4 years in Europe [43]. As a consequence, a migrant population may be more likely to develop schistosomiasis-related chronic complications because it enjoys a longer lifespan. Therefore, the costs associated with a WW strategy—and the benefits of a PT or TT strategy—would increase. On the other hand, it could be argued that better living conditions (i.e. improved nutrition) could avert or slowdown the development of such complications. Other evolving factors could be a shift in the gender balance of migrant populations and a decrease in the mean age through new arrivals. For instance, a younger population might result in an increasing PZQ treatment effectiveness and more long-term associated costs averted, as suggested above.

7. *The cohort effect.* After a certain period of implementation, it might be expected that a screening program would tend to reach more recently arrived migrants and therefore a higher proportion of acute or recent infections would be detected, whereas chronic infections would already have been screened and treated appropriately. As a result, the general prevalence of schistosomiasis would decrease whereas the averted costs per infected-treated person would increase.

**Table 3. Potentially intervening factors, probable effects and recommended actions related to the implementation of schistosomiasis screening program in a non-endemic country.**

| Factor | Probable effect | Recomended research or evaluation action |
|---|---|---|
| Prevalence of different *Schistosoma* species | Differential prevalence of *Schistosoma*-associated morbidities (genitourinary vs. digestive) | Develop more specific diagnostic tools (e.g. PCR-based) |
| PZQ efficacy in long-term residents with chronic infection | Limited efficacy of PZQ in preventing morbidities; increase in the number of doses and dosage for long-term infections | Determine the efficacy of PZQ, including recommended dosages, in long-term migrants with morbidities; ascertain the natural history of schistosomiasis infection in non-endemic countries |
| Coverage of screening program | Limited coverage of exposed population and hard-to-reach pockets; limit coverage of population at high risk (e.g. new and undocumented migrants) | Include costs of outreach and awareness-raising activities in cost analysis; develop POC tests |
| Limited specificity and sensitivity of current diagnostic tests | Increase in the number of unnecessary treatments; false negative results which will result on missed treatment opportunities | Develop more specific and sensitive diagnostic tools, both laboratory and clinical |
| Secondary health-related costs | Increase in associated costs of WW | Include additional secondary costs in the cost analysis |
| Changes in socio-demographic features of target population | Modulation of associated costs of ST, TT and WW depending on age structure, gender distribution, country of origin and legal status | Monitor exposed population and re-assess cost analysis and screening strategies |
| Expanded coverage of schistosomiasis control programs in country of origin | Decrease in prevalence of schistosomiasis among newly arrived migrants | Regularly re-assess prevalence of infection/associated morbidities and the cost analysis of screening strategies |
| Cohort effect of screening programs | Relative increase in number of recent infections, general decrease in prevalence of long-term schistosomiasis infection | Regularly re-assess of cost analysis and screening strategies including prevalence of long-term infection |

8. *Impact of mass treatment programs in endemic countries*. Since 2006, the WHO has recommended the implementation in endemic countries of empirical treatment by mass drug administration of PZQ in school-aged children (<5 years old). New guidelines have expanded this recommendation to children as young as 2 years, lowering the prevalence threshold for annual administration and increasing frequency of treatment [6]. These programs have had a significant impact on incidence and prevalence of the infection in places with good and sustained coverage, with an overall decrease in the 2015–2019 period in sub-Saharan Africa countries in the prevalence of infection by *S. haematobium* of 67·9% (CI 64·6–71·1) and by *S. mansoni* of 53·6% (CI 45·2–58·3) relative to the 2000–2010 period [44]. However, overall coverage in sub-Saharan Africa was limited when most of the exposed individuals in our study were still living there, with preventive chemotherapy reaching less than 14% of the targeted population in 2012 [45]. However, in terms of our projections, if the baseline prevalence of infection is reduced in the countries of origin, this could sharply reduce the prevalence of active schistosomiasis in newly arrived migrants.

In Table 3 we summarize these factors, their possible impact and the related research that could be pursued in order to be better shape both a screening program and a cost analysis.

## Discussion

Under the most likely scenarios (prevalence of infection 20% to 50%), our analysis supports the TT strategy as the most cost-effective policy to prevent schistosomiasis-associated long-term complications in our study population and probably in any EU country with substantial migrant population pockets from schistosomiasis-endemic countries. The main strength of our model is the consideration of a large and detailed range of associated costs using data from our local health system. In spite of the gaps in knowledge that we have identified and the simplicity of our cost analysis, the large potentially averted-costs margins suggest a net benefit in the scenarios we forecast (for instance, 24% prevalence, 90% treatment efficacy and associated

care costs of 70% would result in a net saving of around €25 million). In fact, we did not consider the disability-adjusted life-year associated with long-term morbidities that could be avoided with prompt schistosomiasis treatment, indirect costs like the loss of labor hours, or some very high associated costs like kidney transplant, which can be as high as €40,000. This would increase the benefit of any screening strategy. Our results are in agreement of a recently work published in Italy [46]. However, the results from this previous work and this present article are based on the assumption of a highly sensitive and specific screening tool, which is not the case for the current serology-methods. Serology test is the recommended screening method in non-endemic countries [10], but the value to detect active infections needs to be better ascertained. The lack of sensitivity/specificity of serology test for the TT strategy could be overcome by the inclusion of a check list of signs and symptoms as we suggested previously [13] or three steps combining serology, microscopy examination and ultrasound or clinical assessment [17] or antigen-detection. However, the inclusion of urine/stool examination proposed by the Australian health authorities [17] have been designed for recent arrived migrants, but not for long-term migrants. Otherwise Marchese et al. found a 92% sero-prevalence and 38% of microscopy diagnostic among 272 diagnosed patients [47] and Salas-Coronas found a prevalence of 10.5% among exposed individuals [12], but the time since exposure was lower than one year and around 3 years, respectively. Consequently, the development of new screening tools is one of the main research issues to be addressed for diagnostic of Schistosoma and schistosomiasis-related pathology in non-endemic countries among long-term migrants. A pre-post treatment analysis could help to better ascertain the value of the current serology methods to detect patients which may benefit of PZQ treatment.

Recent studies and guidelines concluded that a PT strategy would provide the best cost-benefit ratio [20, 40]. However, this difference in conclusions could be because those studies focused on recently arrived migrants, who may have more recent or acute infections, whilst we considered long-term residents with a presumably high prevalence of chronic infection. The later population represents the most common profile of a schistosomiasis-exposed population in Europe (in our study setting, most migrants have more than 10 years of residence [13]). However, our estimated costs and averted costs for PT compared to TT strategies are quite close in scenarios of high prevalence (see Figs 1 and 2). These contrasting assessments underscore the unknowns we have identified regarding the natural history of *Schistosoma* infection in non-endemic countries (for instance, the efficacy of PZQ for long-term infections), which have not been taken into account in previous works and the particular clinical profile of long-term exposed migrants [20, 46]. This is why different scenarios of treatment efficacy within reasonable intervals are used in this analysis. In fact, under a situation of high prevalence or recent infections (as is the case of school-aged children in endemic countries), the cost-benefit ratio for a PT strategy greatly improves, and it may be a suitable strategy among the population subset consisting of newly arrived migrants, as suggested by Webb et al. [20]. However, the use of a PT strategy with our targeted population is at the very least controversial. From a medical perspective, a TT strategy would allow the detection and early treatment of schistosomiasis-associated morbidities and reduce the severity of such conditions. With this perspective, the TT strategy including a check list of signs and symptoms would need a referral and linkage to care system for patients suspected of serious morbidities which need a closer follow-up. A Spanish series found a 4% prevalence of severe morbidity among patients with confirmed infection [12], which may increase over time given the chronic nature of the infection. Hospitalization and chronic symptoms have been found highly prevalent among imported schistosomiasis infections in Europe [48]. From a feasibility perspective, the success of a PT strategy depends on its covering a large population, which is unlikely in the foreseeable future in our context. Otherwise, the rationale for mass drug administration strategies is not just to prevent

**Table 4. For and against each strategy according to different considered assumptions.**

| Strategy | For | Against |
|---|---|---|
| **Watchful and waiting strategy** | Lack of efficacy of Schistosomiasis treatment over long-term conditions | PZQ Treatment at early stages of a given chronic complication may result in a better prognostic |
| | Low prevalence of the infection | |
| **Presumptive treatment** | Easier access to hard-to-reach populations at risk | No specific diagnostic would be made |
| | Lack of sensitive and specific diagnostic tools | Low coverage |
| | Access to hard-to-reach populations | Lack of follow-up of patients |
| | High effective in recently arrived migrants from endemic countries | Risk of a high number of unnecessary treatments with secondary effects |
| | High prevalence of infection | |
| **Test and treat** | More specific and targeted treatment | Lack of specific and sensitive diagnostic tests and screening procedures |
| | Availability of sensitive and specific diagnostic tools | |
| | Allows to follow-up patients and evaluate treatment efficacy | |
| | High-middle prevalence of the infection | Low coverage |

infection, but also to interrupt transmission and infectiousness, which is not the primary issue in non-endemic countries, even though limited local schistosomiasis transmission and outbreaks in Europe have been described recently [49–51]. Finally, the high rate of lost-of-follow-up of patients under schistosomiasis-infection evaluation observed in our study region [13] and other sites [12], supports a community-based catch and treat option (PT or TT) even with a single dosage to ensure the effective PZQ intake. With this regard, a point-of-care (POC) would be most appropriated for a TT strategy. In Table 4 we show the factors for and against PT, TT or WW strategies.

A research agenda should be developed on the basis of the gaps in knowledge identified here (see Tables 3 and 4). The determination of the cut-off efficacy of PZQ (including the number of doses) in chronic infections or long-term residents in Europe is a key question, since this information would facilitate estimation cost of the potential long-term morbidities that could be averted and the resulting cost-benefit ratio. Future research should also focus on the development of POC tests (e.g. a finger-prick blood test) [52] that would help to effectively implement a rapid test-and-treat strategy, and the development of more specific diagnostic methods in order to avoid unnecessary treatment (e.g. a PCR-based diagnostic that could differentiate among *Schistosoma* species, for clinical and/or laboratory use) to overcome the limitations of the serology test (including false-negative results). Research progress in this area has recently been reviewed in a recent work [53]. The development of a POC strategy would greatly reduce the associated costs of the TT strategy with a single visit and cheaper test, which would be even lower if only one dose of PZQ were administered. On a theoretical level, a POC-test which halves the costs of the serology we used (€24), one visit and a single dose of PZQ would result in a 30% cost decrease compared to a TT-2 strategy in a 24% prevalence scenario and would probably increase the feasibility of the program. Additionally, the evolving socio-demographic characteristics of the targeted population point to the need for regular assessment of the cost analysis and, if necessary, the reshaping of screening strategies, which should include secondary costs associated with a WW approach and complementary outreach activities. Regarding the coverage of mass treatment programs in the countries of origin, schistosomiasis prevalence in sub-Saharan Africa has decreased considerably, most likely because of the scale-up of preventive chemotherapy [44]. For the moment, the impact of this factor on our targeted population is still limited, since they are mainly long-term residents, but the

prevalence of schistosomiasis may decrease in non-endemic countries in the near future as a result of these programs.

A range of factors such as time and degree of exposure to *Schistosoma*, age, gender and country of origin can modify the risk of contracting the disease [54]. Consequently, a more comprehensive cost analysis should consider different subsets of exposed populations, each of which may require a tailored intervention and associated use of resources. Such population subset studies might look at long-term resident migrants vs. recent migrants, infants, females, specific countries of origin, geographic regions or even villages or second generation migrants born in Europe that travel regularly to their family's country of origin. The PT strategy might be the most optimal option in a specific subset of these populations (e.g. those with more recent infections and highest prevalence). Such studies could also explore the issue of undocumented migrants, who have not been included in our estimates and are an especially hard-to-reach population. With this regard, success depends on a high degree of compliance in the taking of medication, and therefore a single-dose strategy (PT or TT-1) with an on-site intake of PZQ once a patient has been reached (PT) or diagnosed (TT) is likely to offer better effectiveness in the real world. With this regard, acceptability studies seems to be most appropriated prior the implementation of a screening program.

In conclusion, the TT strategy might be the most cost-optimum strategy for long-term migrants in Europe, but the study underscores as well a number of key questions and research gaps related to imported schistosomiasis in non-endemic countries. For instance, we were unable to determine with any precision the cut-off point for costs associated with long-term morbidity treatment, or those related to PZQ efficacy. These are the main limitations of our analysis, and as a result our estimates are necessarily discretional. In the meantime, these significant gaps in our knowledge serve to confirm the perception of schistosomiasis in non-endemic countries as a neglected disease. In this regard, it is important to underline that the proposed research agenda would certainly also have positive implications for schistosomiasis-endemic countries, which would benefit from the development, for example, of new POC methodologies or a clearer understanding of PZQ efficacy in long-term exposed patients. The importance of these repercussions is clear, given that the vast majority of schistosomiasis infections and related morbidities will still occur in those countries.

## Supporting information

**S1 Text. Supplementary tables and figure. Table A in S1 Text**: Mean expected estimated cost for each schistosoma-associated pathology and occurrence proportion. **Tables B in S1 Text**: Estimated averted costs with TT strategy with 2 PZQ dosage. In the y axis the estimated rate of cure/estimated prevalence of the infection in the targeted population. Shadowed in grey we emphasize the scenarios with a negative results. Averted costs are expressed in Euros. **Table C in S1 Text:** Estimated averted costs with TT strategy with 2 PZQ dosage. In the y axis the estimated rate of cure/estimated prevalence of the infection in the targeted population. Shadowed in grey we emphasize the scenarios with a negative results. Averted costs are expressed in Euros. **Table D in S1 Text:** Estimated averted costs with PT strategy with 1 PZQ dosage. In the y axis the estimated rate of cure/estimated prevalence of the infection in the targeted population. Shadowed in grey we emphasize the scenarios with a negative results. Averted costs are expressed in Euros. **Fig A in S1 Text**: Theoretical averted costs under different scenarios of Schistosoma prevalence, long-term morbidities-associated costs and treatment efficacy for PT and TT strategies and WW strategy*. *Dashed horizontal line indicates the zero balance. (DOCX)

**S1 Data. Dynamic table of schistosomiasis screening cost-analysis.**
(XLSX)

## Author Contributions

**Conceptualization:** Sílvia Roure, Francesc López, Olga Pérez-Quílez, Xavier Vallès.

**Data curation:** Francesc López, Irene Oliva, Oriol March, Anna Chamorro, Xavier Vallès.

**Formal analysis:** Sílvia Roure, Francesc López, Irene Oliva, Oriol March, Anna Chamorro, Xavier Vallès.

**Investigation:** Sílvia Roure, Francesc López, Israel López Muñoz, Amaia Castillo.

**Methodology:** Sílvia Roure, Francesc López, Lluís Valerio, Xavier Vallès.

**Project administration:** Francesc López, Olga Pérez-Quílez.

**Resources:** Elena Abad.

**Supervision:** Sílvia Roure, Francesc López.

**Validation:** Francesc López, Laura Soldevila, Manolo Lozano, Helena Masnou, Mario Oliveira, Laura Cañas, Mireia Gibrat, Marta Chuecos, Juan José Montero, Karen Colmenares, Gemma Falguera, Josep Maria Bonet, Mar Isnard, Núria Prat, Oriol Estrada, Bonaventura Clotet.

**Writing – original draft:** Sílvia Roure, Francesc López, Xavier Vallès..

**Writing – review & editing:** Sílvia Roure, Francesc López, Irene Oliva, Olga Pérez-Quílez, Oriol March, Anna Chamorro, Elena Abad, Israel López Muñoz, Amaia Castillo, Laura Soldevila, Lluís Valerio, Manolo Lozano, Helena Masnou, Mario Oliveira, Laura Cañas, Mireia Gibrat, Marta Chuecos, Juan José Montero, Karen Colmenares, Gemma Falguera, Josep Maria Bonet, Mar Isnard, Núria Prat, Oriol Estrada, Bonaventura Clotet, Xavier Vallès.

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
