## [Decision Letter · Decision Letter 0]

28 Sep 2022

Dear Vallès,

Thank you very much for submitting your manuscript "Schistosomiasis screening in non-endemic countries from a cost perspective: knowledge gaps and research priorities" for consideration at PLOS Neglected Tropical Diseases. As with all papers reviewed by the journal, your manuscript was reviewed by members of the editorial board and by several independent reviewers. In light of the reviews (below this email), we would like to invite the resubmission of a significantly-revised version that addresses all reviewer comments and suggestions. 

We cannot make any decision about publication until we have seen the revised manuscript and your response to the reviewers' comments. Your revised manuscript is also likely to be sent to reviewers for further evaluation.

Sincerely,

Suzy J Campbell, PhD

Section Editor

Suzy Campbell

Section Editor

Reviewer's Responses to Questions

**Key Review Criteria Required for Acceptance?**

**Methods**

-Are the objectives of the study clearly articulated with a clear testable hypothesis stated?

-Is the study design appropriate to address the stated objectives?

-Is the population clearly described and appropriate for the hypothesis being tested?

-Is the sample size sufficient to ensure adequate power to address the hypothesis being tested?

-Were correct statistical analysis used to support conclusions?

-Are there concerns about ethical or regulatory requirements being met?

Reviewer #1: -The objectives are to compare theoretically 3 strategies and their cost against imported schistosomiasis (spontaneous evolution, presumptive teatment, test and treat if +); to look at factors influencing the benefits of each strategy. Assumptions set-up for the decision analysis model are presented.

It is an interdisciplinary study with a valuable insight in the approaches of schistosomiasis management in a South european country, one of the main "entry gate" of Sub-Saharan immigrants in Europe.

-The question of serious morbidity detection in the management of Schistosomiasis screening appears understated.

-The population described live in a sizable single area which welcomes a high number of immigrants and where concentrate primary care facilities: 74 000 people estimated to be exposed to chronic infection. However, socio-demographic characteristics are not detailed.

-Authors present with honesty the numerous knowledge gaps that limit significantly the results, especially the factors influencing each strategy. 

-No ethical concern.

Reviewer #2: (No Response)

**Results**

-Does the analysis presented match the analysis plan?

-Are the results clearly and completely presented?

-Are the figures (Tables, Images) of sufficient quality for clarity?

Reviewer #1: The analysis plan lack of solid assumptions on certain points: prevalence of infection, prevalence of serious morbidity, screening method.

Authors could have presented a Table listing the assumptions in each strategy with their respective evidences FOR and AGAINST them. 

Although it is specified that results apply to the local situation of Barcelona, some of the assumptions are drawn not from local or national studies, even though such data exist. Concerning the prevalence of infection, it is estimated by a pooled seroprevalence from international surveys, some conducted quite a long time ago (Asundi et al.). Authors could have extracted from this paper Spanish or South-European study data reviewed, in order to get a sharper figure. 

Indeed, infection prevalence (not seroprevalence) depends closely on the sociodemographic profile of the immigrants in the place they settled: age, sex, geographic origin... at the village level, because of the overdispersed distribution of Schistosomes among populations and age groups.

The other inconvenient is that it is a seroprevalence which overestimates the real situation. The fact that it corresponds to Schisto prevalences (mostly based on microscopic examination) in the six African countries most represented in the immigrant community of Barcelona seemed to be a coincidence. 

In the first scenario (Watchfull waiting), assumptions concerning the organ involvements in Schistosomiasis (serious morbidity) are rare, although the primary issue of the study is to prevent or reduce this occurence after it has been detected. In this light, a body of a dozen publications on the subject in non endemic European countries are lacking as references: about 4 are Spanish, 4 or 6 are from Italy, 2 or 3 come from France. About the rate of hospitalization mentioned in Table 2, the EurotropNet surveillance data

reported a 11 % rate among migrants diagnosed with positive urine microscopy (Lingscheid et al.).

As for assumptions, estimated frequency of symptoms of infection, s. of complication, relative part of Schisto species among the popultion studied (about 80 S. haem./20 S. mansoni ?), estimated effect of praziquantel on regression of organ involvement, accessibility of migrants to primary care facilities, estimated under-diagnosis in these settings, etc.

In fact, authors chose to present some of these latters among "Co-factors that may influence the cost-benefit ratio" (p. 20). The choice is respectful but some should appear before, among the assumptions presentation, the 2 most important in the Watchfull Waiting scenario being that: 

-ultrasonography is the gold standard to detect organ involvements

 -ultrasonography abnormalities considered to be related to such involvements are detected significantly more often in case of positive microscopic examination of urine than in case of positive serology. It seems it has not been assessed for intestnal schistosomiasis. 

An recent Italian paper presenting evidences of regression of bladder involvements concludes "Mucosal masses in young patientsregressed after treatment without recurrence, supporting the recommendation that invasive procedures should be avoided unless lesions or other signs of concern persist for > 6 months."

The second scenario (Presumptive treatment) could be assumpted not only by studies on praziquantel efficacy, but acceptability studies of such administration. People usually prefer to take a medicine for an identified cause rather than "in case of ...". One important assumption of this scenario is the number of doses of praziqantel. If we empirically think that 2 would be more efficient than one, at which time interval should they be delivered ? How many persons with organ involvement would not be cured enough with 2 doses ?.. an ethical question raised by this scenario. 

The third scenario (Test and treat) is based on serology testing. As previously mentioned, positive serology is not significantly associated with organ involvements. Moreover, it can give false negative results, an outcome representing a loss of chance in matter of health to people in this case with organ involvement. In addition it is not as relevant in detecting Schisto infection in a population group where prevalence of infection is high. As we don't know precisely the prevalence in Barcelona, the choice may be risky. Unfortunately authors don't suggest alternatives to serology that could be assessed. It is a pity that microcopic examination is not mentioned in the paper, even if it is not a good screening method, theoretically. Some papers (not mentioned in the references) explore the feasibility of screening procedures using clinical checklist and serology, or a step by step clinical assessment with urine dipstick in the detection of S. haematobium. The Australian health Authorities foresee a parasitic examination of stools and urine among migrans from endemic regions, after a positive serology, alongside the cure.

These "two step" procedures may be worth to be assessed, even as screening procedures.

Reviewer #2: (No Response)

**Conclusions**

-Are the conclusions supported by the data presented?

-Are the limitations of analysis clearly described?

-Do the authors discuss how these data can be helpful to advance our understanding of the topic under study?

-Is public health relevance addressed?

Reviewer #1: More than the fact that the Test and treat strategy seems to be the best one in the population studies according to authors, they appropriately stress on the big gap of lack of knowledge about Schisto in non endemic areas. The limitations of analysis is clearly described (in the paragraph "Co-factors that may influence the cost-benefit"). Indeed, they propose a research agenda of further studies, on treatment efficacy, POC tests, etc., in the light of public health challenges.

Reviewer #2: (No Response)

**Editorial and Data Presentation Modifications?**

Reviewer #1: A table comparing the results of the 3 strategies could help the reader at the beginning of the paper.

Table 1: please precise that prevalence figures presented are mostly based on parasitic microscopic examinations (to check if possible in the referenced article)

Tables 2 and 3 could be re-arranged for better understanding: 

-Table 2 could list Symptoms only (the more specific, if possible) and associated diagnosis procedures (instead of "care ..."). Symptoms could be grouped in genitourinary (S. haematobium), intestinal (S. mansoni) and neurologic Schisto. The diagnosis procedures could be inspired by recommendations in clinical practice issued from the Spanish Council of Medicine or its equivalent.

All the diagnosis procedures listed do not seem necessary to appear, but the most important are Ultra-sonography and CT.

-Table 3: could retain only the pathologies (take off "blood in stools", "hematuria" and other symptoms), which could be hierachied in order of frequency. The estimated occurrence is not referenced for "Female infertility", "stroke" and "pulmonary hypertension". Authors should state that some of these pathologies cannot be ruled out for sure as Schistosomiasis complications. besides to the weighted cost, it could be valuable to add the "Mortality rate" (if we have this data in non endemic countries, unless it should be specified).

table 4: Column "Probable effects", please complete for some factors; column "Recommended research action", please focus on the priority aim of your study i.e. to prevent or detect schisto complications (organ pathologies).

References

Relatively important references are lacking, as written above, on:

 Complications: Tamarozzi et al. and other European studies

Prevalence of infection in European countries: Marchese et al.

strategy of screening: ....

Reviewer #2: (No Response)

**Summary and General Comments**

Reviewer #1: First I have to say that I am not a Health economist and my scientific background is therefore partially relevant to the subject of the paper. I could not expertise the calculation of the cost estimates and hope that other reviewers will do it. However, some assumptions made for building the various scenarii, especially the first one which acts as basis (spontaneous evolution) have raised my interest. 

This paper do not raise the question of a global health burden in the host country, but it opens the persepctive of Schisto screening being a part of an extended health assessment for migrants.

Reviewer #2: This paper investigates an important reach gap regarding the costs of schistosomiasis screening programs in non-endemic countries. I have the following comments/suggestions

Major comments 

I understand that the authors have looked at a lot of scenarios. However, overall I found that results very hard to follow. In particularly the averted costs need to be made much clear – potential have results tables rather than just figures (even if some of the scenarios are just reported in the appendix). It needs to be made clearer what is driving these averted costs.

Results to the above expand on the thresholds were the optimum intervention changes 

The Estimated Proportion of occurrence (%) within Table 3 needs to have a reference number that corresponds to the reference list. In the legend it just says WHO for most of these.

I do not see how a treatment efficiency of 100% could be reasonable. Do you have any references to support this

What assumption were made regarding the diagnostic sensitivity 

In many places you talk about the “cost/benefit ratio” – I believe this is confusing as my understanding is that this is meant to be a costing study and not a cost-benefit analysis? If it is meant to be a economic evaluation (i.e cost-benefit analysis) the methods needs to be expanded upon. It needs to be made clearer what criteria you are using to say an strategy is optimum. 

I found the Costs associated with PT and TT strategies hard to follow and suggestion these are summarized in a table. Related to this a Table summarizing the different strategies could be helpful

Minor comments 

In many places you talk about the “cost/benefit ratio” – I believe this is confusing as my understanding is that this is meant to be a costing study and not a cost-benefit analysis.

Abstract; I found the following sentence hard to understand – consider rephasing: “The difference in averted costs between test-and-treat and watchful waiting strategies ranges from nearly €60 million in scenarios of 30% prevalence, 100% associated costs and 100% treatment efficacy, to a neutral cost/benefit ratio when these parameters are halved.”

Table 3: Some of the values seem to have 4 decimal places – would consider rounding.

I would consider not having abbreviations for the different strategies. It may save words but makes it much harder to read (this is just my opinion and is the authors decision)

PLOS authors have the option to publish the peer review history of their article (what does this mean?). If published, this will include your full peer review and any attached files.

Reviewer #1: No

Reviewer #2: No
---

## [Decision Letter · Decision Letter 1]

8 Feb 2023

Dear Vallès,

Thank you very much for submitting your manuscript "Schistosomiasis screening in non-endemic countries from a cost perspective: knowledge gaps and research priorities" for consideration at PLOS Neglected Tropical Diseases. As with all papers reviewed by the journal, your manuscript was reviewed by members of the editorial board and by several independent reviewers. The reviewers appreciated the attention to an important topic. Based on the reviews, we are likely to accept this manuscript for publication, providing that you modify the manuscript according to the review recommendations. 

Sincerely,

Suzy J Campbell, PhD

Section Editor

Suzy Campbell

Section Editor

Reviewer's Responses to Questions

**Key Review Criteria Required for Acceptance?**

**Methods**

-Are the objectives of the study clearly articulated with a clear testable hypothesis stated?

-Is the study design appropriate to address the stated objectives?

-Is the population clearly described and appropriate for the hypothesis being tested?

-Is the sample size sufficient to ensure adequate power to address the hypothesis being tested?

-Were correct statistical analysis used to support conclusions?

-Are there concerns about ethical or regulatory requirements being met?

Reviewer #1: same as in the first review

Reviewer #2: (No Response)

**Results**

-Does the analysis presented match the analysis plan?

-Are the results clearly and completely presented?

-Are the figures (Tables, Images) of sufficient quality for clarity?

Reviewer #1: same as in the first review

Reviewer #2: (No Response)

**Conclusions**

-Are the conclusions supported by the data presented?

-Are the limitations of analysis clearly described?

-Do the authors discuss how these data can be helpful to advance our understanding of the topic under study?

-Is public health relevance addressed?

Reviewer #1: same as in the first review

Reviewer #2: (No Response)

**Editorial and Data Presentation Modifications?**

Reviewer #1: YES : to add a sub-tittle to the article’s title : «Schistosomiasis screening in non-endemic countries from a cost perspective : knowledge gaps and research priorities. The case of African long-term residents in the Metropolitan Area, Spain. »

I suggest to add the same precision in the Objectives of the Abstract

Reviewer #2: (No Response)

**Summary and General Comments**

Reviewer #1: I read the answers of the Authors.

I agree with most of the modifications they did.

I agree with the stress made by the authors on the characteristic of the migrant population studied, i.e. the fact that more than half have been living there for more than 10 years .

I therefore suggest to add a sub-tittle to the article’s title : «Schistosomiasis screening in non-endemic countries from a cost perspective : knowledge gaps and research priorities ».The case of African long-term residents in the Metropolitan Area, Spain.

I suggest to add the same precision in the Objectives of the Abstract.

Reviewer #2: The authors have comprehensively addressed my comments. I have the flowing minor comments

There are a number of small typos (particularly around million), extra “.” and “,” and errors in the bullet point numbering (line 476. The manuscript needs another in-depth proofread.

Avoid saying “cost-effective strategy” in the conclusions– as you have not done a cost-effectiveness analysis. Maybe say cost-optimum 

There are a number of small typos (particularly around million), extra “.” and “,” and errors in the bullet point numbering (line 476. The manuscript needs another in-depth proofread.

I would avoid having gridline in the fingers – for figure 2 it is hard to tell which line is a strategy vs a gridline

“Not all morbidity-associated care procedures are indicated for any given patient (the 100%-cost scenario” I found this unclear. Please can you clarify more clearly what the % Scenarios for the WW strategy and morbidities-associated costs are (i.e. what is meant by 100% of the costs being incurred here). A table that summarizes the strategies/ scenarios that corresponds to the main figures might be helpful – particularly as the treatment efficacy range and treatment costs for long-term morbidities range are both 40%-100%.

What are the treatment efficacy assumptions within the two main figures?

Add abbreviations to the figure legends.

“The neutral balance (zero averted costs)” I think saying zero averted costs here might confuse the readers – maybe say it is cost neutral. 

Make sure the terms are consistent. For example, in Table 4 it says Wait and watchful strategy vs watchful waiting strategy in the text

PLOS authors have the option to publish the peer review history of their article (what does this mean?). If published, this will include your full peer review and any attached files.

Reviewer #1: Yes: François Deniaud MD

Reviewer #2: No

Figure Files:

Data Requirements:

Reproducibility:

References

---

## [Decision Letter · Decision Letter 2]

7 Mar 2023

Dear Vallès,

We are pleased to inform you that your manuscript 'Schistosomiasis screening in non-endemic countries from a cost perspective: knowledge gaps and research priorities.The case of African long-term residents in the Metropolitan Area, Spain' has been provisionally accepted for publication in PLOS Neglected Tropical Diseases.

Best regards,

Francesca Tamarozzi

Section Editor

Francesca Tamarozzi

Section Editor

Reviewer's Responses to Questions

**Key Review Criteria Required for Acceptance?**

**Methods**

-Are the objectives of the study clearly articulated with a clear testable hypothesis stated?

-Is the study design appropriate to address the stated objectives?

-Is the population clearly described and appropriate for the hypothesis being tested?

-Is the sample size sufficient to ensure adequate power to address the hypothesis being tested?

-Were correct statistical analysis used to support conclusions?

-Are there concerns about ethical or regulatory requirements being met?

Reviewer #2: (No Response)

**Results**

-Does the analysis presented match the analysis plan?

-Are the results clearly and completely presented?

-Are the figures (Tables, Images) of sufficient quality for clarity?

Reviewer #2: (No Response)

**Conclusions**

-Are the conclusions supported by the data presented?

-Are the limitations of analysis clearly described?

-Do the authors discuss how these data can be helpful to advance our understanding of the topic under study?

-Is public health relevance addressed?

Reviewer #2: (No Response)

**Editorial and Data Presentation Modifications?**

Reviewer #2: (No Response)

**Summary and General Comments**

Reviewer #2: (No Response)

PLOS authors have the option to publish the peer review history of their article (what does this mean?). If published, this will include your full peer review and any attached files.

Reviewer #2: No

---

## [Editor Report · Acceptance letter]

30 Mar 2023

Dear Vallès,

We are delighted to inform you that your manuscript, "Schistosomiasis screening in non-endemic countries from a cost perspective: knowledge gaps and research priorities.The case of African long-term residents in a Metropolitan Area, Spain," has been formally accepted for publication in PLOS Neglected Tropical Diseases.

Best regards,

Shaden Kamhawi

co-Editor-in-Chief

Paul Brindley

co-Editor-in-Chief
